# Identification of miRNAs as Biomarkers of Cardiac Protection in Non-Genetically Modified Primary Human Cardiomyocytes Exposed to Halogenated Hypnotics in an In Vitro Model of Transfection and Ischemia/Reperfusion: A New Model in Translational Anesthesia

**DOI:** 10.3390/life13010064

**Published:** 2022-12-25

**Authors:** Maria Dolores Carmona-Luque, Laura Gonzalez-Alvarez, José Luis Guerrero Orriach

**Affiliations:** 1Maimonides Institute of Biomedical Research in Cordoba (IMIBIC), 14004 Cordoba, Spain; 2Cellular Therapy Unit, Reina Sofia University Hospital, 14004 Cordoba, Spain; 3Institute of Biomedical Research in Malaga, 29010 Malaga, Spain; 4Department of Anesthesiology, Virgen de la Victoria University Hospital, 29010 Malaga, Spain; 5Department of Pharmacology and Pediatrics, School of Medicine, University of Malaga, 29010 Malaga, Spain

**Keywords:** cardioprotection, anesthesia, human cardiomyocytes, sevoflurane, propofol

## Abstract

Background: Many clinical studies have identified some circulating micro-RNAs (miRNAs) as potential biomarkers with regard to the cardioprotective effects of halogenated agents administered perioperatively during myocardial conditioning procedures. However, there is a major methodological difficulty in identifying these potential miRNA targets in cardiac cells. Methods: We developed an in vitro protocol to analyze the differential expression of target miRNAs at the intracellular level in non-genetically modified primary human cardiomyocytes (HCMs) through their exposure to different hypnotic compounds (i.e., halogenated versus non-halogenated). For this purpose, we performed a validated in vitro model of “ischemia and reperfusion” with the transfection of specific miRNA mimics (MIMICs) designed to simulate naturally occurring mature miRNAs as a functional study. Afterwards, next-generation sequencing (NGS) was used to identify and quantify miRNAs and elucidate their function. The differences in miRNA expression between HCMs exposed to different hypnotic drugs, along with the prediction of functional miRNA targets, were assessed using a meticulous in-house bioinformatics pipeline in order to derive diagnostic biomarkers and possible therapeutic targets. Conclusion: In brief, this methodological procedure was designed to investigate whether the cardioprotective effects of halogenated agents are a phenomenon mediated by either the activation or the suppression of miRNAs targeted by halogenated anesthetics.

## 1. Introduction

The number of patients diagnosed with heart failure is increasing and projected to rise by 46% by 2030, resulting in more than 8 million people with heart failure [1]. Most of these patients with heart damage will be candidates for myocardial revascularization surgeries and, therefore, undergoing anesthesia. However, surgical myocardial revascularization—both intraoperatively and postoperatively—gives rise to ischemic events in a high number of patients.

Currently, there are no predictive markers of morbidity that can be used to detect patients with a high risk of suffering perioperative ischemia or arrhythmias at baseline prior to the intervention [2]. At the anesthetic level, an advance in the treatment of this group of patients is the knowledge of the cardioprotective properties of halogenated hypnotic drugs [3], described for the first time in 1986 [4].

The cardioprotective effects of halogenated agents (in comparison to intravenous anesthetics) administered perioperatively in cardiac surgery depend on exposure time. Thus, longer exposure to halogenated agents at the standard concentrations routinely used in clinical practice exerts greater clinical benefits [5].

Myocardial conditioning has mostly been analyzed in patients undergoing surgical myocardial revascularization, and the mechanisms by which cardioprotection is elicited with exposure to halogenated agents throughout the ischemia/reperfusion intervention have been described [6].

Myocardial pre- and postconditioning are triggered by the activation of several enzymatic pathways. Some of the proteins involved in these pathways have been documented as being overexpressed in cardiac surgery patients, and a differential increase has been observed regarding halogenated agents administered intra- and postoperatively as hypnotic drugs [5].

Micro-RNAs (miRNAs) are small non-coding RNA molecules that regulate gene expression at the post-transcriptional level by silencing or degrading messenger RNAs (mRNAs). Moreover, the miRNAs are involved in regulating various biological processes, such as cell differentiation, proliferation, apoptosis induction, and embryonic and tissue development. Recently, miRNAs have been a major focus of interest in clinical research regarding the identification of new therapeutic miRNA biomarkers that could be candidates for different pathologies, including cardiovascular diseases [3].

It has been shown that many miRNAs are involved in the damage induced by ischemia/reperfusion (I/R) procedures in the heart, and evidence indicates that miRNAs could play an important role in cardiac protection and heart failure [7]. Moreover, some circulating miRNAs have been identified as potential biomarkers regarding the cardioprotective effects of halogenated agents administered perioperatively during myocardial conditioning procedures. Recently, our group has published preliminary results that show several circulating miRNAs as mediators of cardioprotection in patients who received sevoflurane as a halogenated agent during cardiac surgery. Differences in the expression of miRNAs associated with better prognosis of ischemic heart disease have been observed in patients treated with halogenated drugs. This differential expression of miRNAs has been associated with the activation of mediators of anesthetic-induced pre- and postconditioning associated with reduced cell apoptosis and with decreased caspase and TNF-alpha concentrations [5].

However, in most published studies, the miRNAs have been analyzed in patients at the plasma level as circulating miRNAs, or in genetically modified cardiomyocytes.

Therefore, the aim of the present work was to develop a methodological procedure that allows the identification and validation of these potential miRNA biomarkers at the intracellular level in primary human cardiomyocytes isolated from normal human ventricular tissue of the adult heart (PromoCell^®^, Heidelberg, Germany), suitable for clinical application without subjecting the patient to serious surgical risk or genetically modifying non-cardiac cells such as fibroblasts or embryonic stem cells.

The clinical translation of the results obtained after the development of our proposed in vitro procedure is clearly justified by the immediate clinical application of the miRNAs identified as biomarkers of cardiac protection.

## 2. Materials and Methods

A schematic summary of the experimental design is shown in Figure 1. In brief, this protocol required primary human cardiomyocytes (HCMs) to be thawed, cultured under standard conditions (O_2_ 21%, CO_2_ 5%, 37 °C), and transfected (PromoCell^®^, Heidelberg, Germany). Secondly, transfected cells were exposed to hypnotic drugs (e.g., propofol or sevoflurane), adding them to the culture medium. Thirdly, the cells were cultured under anoxic conditions (O_2_ concentration 0%) in a hypoxic chamber to simulate the ischemia process. Next, a reperfusion stage was carried out, culturing the cells under standard conditions. Ischemia and reperfusion processes were performed with HCMs exposed to hypnotic drugs. Once the reperfusion stage was finished, the hypnotic drugs were withdrawn from the culture medium and the cells were cultured under standard conditions for their stabilization.

Finally, to carry out the gene sequencing (NGS), the cells were trypsinized, washed, and conserved at −80 °C as dry pellets. Once the genetic sequencing was completed, a qPCR analysis was performed with the miRNAs identified as significant from the NGS procedure (Figure 1).

All experimental procedures were repeated three times for the validation of the experimental design (n = 3).

### 2.1. Primary Human Cardiomyocytes Culture

This protocol was developed with primary human cardiomyocytes isolated from normal human ventricular tissue of the adult heart acquired from PromoCell^®^; however, it could be adapted to primary human cardiomyocytes isolated from fresh tissue prepared according to the standard procedures described in several published studies [8].

Following the manufacturer’s instructions, a cryovial stored in liquid nitrogen containing thawed HCMs in culture passage 2 was immersed in a water bath at 37 °C for two minutes. After that, the cell suspension was transferred to a culture flask containing prewarmed complete myocyte medium at a seeding density of 15 × 10^3^ cells per cm^2^, and then it was incubated at 37 °C, CO_2_ 5%, O_2_ 21% in a humidified atmosphere for cell attachment. The medium was refreshed after 24 h and every three days until reaching 80% cell confluence.

The manufacturer’s instructions suggest that HCMs must be used before culture passage eight; however, we recommended carrying out this protocol between culture passages four and five, because we observed a detriment in cellular morphology in the last culture passages (Figure 2).

To subculture the cells, we used the detach kit from PromoCell^®^ (Heidelberg, Germany), following the manufacturer’s instructions. In brief, the cell culture supernatant was aspirated, and then we added 100 µL of HEPES solution per cm^2^ of flask surface to wash the attached cells, retired them, and added 100 µL of trypsin solution per cm^2^ of flask surface to detach the cells. Next, we added 200 µL of trypsin neutralization solution per cm^2^ of flask surface, and the cell suspension was aspirated and washed by centrifugation for 3 min at 220× *g*. The supernatant was discarded, and the cells were resuspended in complete culture medium. Cell viability was determined by Trypan Blue Solution staining, and cell counts were performed in a Neubauer chamber to quantify the number of living cells. Finally, the HCMs were reseeded in new flasks at the right planting density.

### 2.2. Transfection Protocol

The miRCURY LNA miRNA Mimics were designed to simulate mature miRNA targets and were transfected into HCMs according to the protocol described below.

In the reviewed literature, no published study described a specific transfection reagent to transfect miRNAs into primary HCMs; however, there are specific transfection reagents to transfect siRNA, siRNA plus plasmids, or DNA into human primary cells.

For this reason, we carried out a fine-tuning protocol to select the best transfection conditions, with the aim of identifying the best transfection reagent concentration to transfect our primary HCMs. Finally, the TransIT-siQUEST^®^ transfection reagent (Mirus Bio Corporation, Madison, WI, USA) was selected. Figure 3 shows the development of the protocol design for this assay. 

Briefly, the transfection protocol was developed as follows: In a 12-well culture plate, different volumes of the aliquoted miRNA Mimic Negative Control-FAM-5′conjugated (66.67 µM) were added per well to identify the optimal working concentration: miC1 = 2.5 nM~5 pmol, miC2 = 5 nM~10 pmol, and miC3 = 5 nM~10 pmol. Moreover, two different concentrations of transfection reagent (TransIT-siQUEST^®^, Mirus Bio Corporation, Madison, WI, USA) were tested for each miRNA Mimic Negative Control concentration: TR1 = 3 µL and TR = 5 µL (Figure 3). Four hours post-transfection, the culture medium was refreshed, and 24 h later the transfection efficiency was analyzed by optical and fluorescence microscopy.

Finally, according to the obtained results, we determined the optimal miRNA mimic concentration to be 5 nM (miC2) and the optimal TransIT-siQUEST^®^ transfection reagent concentration to be 3 µL per mL of culture medium (TR1). These final experimental results are shown in Figure 4.

### 2.3. Ischemia/Reperfusion Protocol Development and Exposure of Cells to Hypnotic Drugs

To carry out the culturing of HCMs under continuous exposure to hypnotic drugs, and to simulate the surgical ischemia/reperfusion process of myocardial conditioning after ischemic damage under exposure to hypnotic drugs, 24 h post-transfection, transfected and non-transfected HCMs were exposed to sevoflurane or propofol as halogenated and non-halogenated hypnotic drugs, respectively, at concentrations similar to clinical concentrations: 280 µM for sevoflurane (similar to 1 minimum alveolar concentration (MAC)), and 1 µM for propofol (equivalent to the median effective dose (ED50) applied in surgical procedures t).

For this purpose, transfected and non-transfected HCMs were cultured in complete medium with sevoflurane (280 µM) or propofol (1 µM) for 12–24 h. After this, the complete medium was replaced by the same volume of Hank’s Balanced Salt Solution (HBSS) supplemented with calcium and magnesium, cytochalasin B (5 µg/mL), and sevoflurane or propofol, maintaining the same concentration of both hypnotics. The cells were incubated for 30 min. Then, the supplemented HBSS solution was replaced with a reduced volume of the HBSS solution that was previously exposed to anoxic conditions. The HCMs were cultured under anoxic conditions in a hypoxia chamber (BioSpherix, Parish, NY, USA) for 90 min to simulate ischemia. After that, the anoxic solution was replaced with a normal volume of non-anoxic supplemented HBSS solution, and the cells were incubated under standard culture conditions for 30 min. This stage was identified as the reperfusion stage. After this incubation, the HBSS solution was replaced with complete culture medium without hypnotic drugs, and the HCMs were incubated for 30 min under standard culture conditions as a stabilization stage for further analysis.

The next step was to carry out the next-generation gene sequencing (NGS) in transfected and non-transfected HCMs. For this purpose, the cells were trypsinized, washed, and conserved at −80 °C as dry pellets (Figure 5).

### 2.4. miRNA Extraction and Qualification

Total RNA with enhanced miRNA enrichment was extracted from the human cardiomyocytes (HCMs) with the RNeasy Plus Universal Kit (Qiagen, Hilden, Germany), following the manufacturer’s recommendations. The aliquot was used to perform an ultrasequencing of miRNA libraries to analyze the miRNA expression pattern. RNA purity was checked using the NanoPhotometer^®^ spectrophotometer (IMPLEN, Westlake Village, CA, USA). Next, the RNA concentration was measured using the Qubit^®^ RNA Assay Kit in the Qubit^®^ 3.0 Fluorometer (Life Technologies, Carlsbad, CA, USA). Finally, RNA integrity was assessed using the RNA Nano 6000 Assay Kit with the Agilent Bioanalyzer 2100 system (Agilent Technologies, St. Clara, CA, USA).

### 2.5. Library Preparation for Small RNA Sequencing

Sequencing libraries were generated using a NEBNext^®^ Multiplex Small RNA Library Prep Set for Illumina^®^ (New England Biolabs (NEB), Ipswich, MA, USA), following the manufacturer’s recommendations [9]. The NEBNext Small RNA kit is designed for generating small RNA libraries. The kit takes advantage of the hallmark 3′ hydroxyl group that miRNAs have as a result of the enzymatic cleavage by Dicer (and other RNA-processing enzymes) and an adenylated 3′ adapter that specifically ligates to this end. By targeting the 3′ end of small RNAs, unwanted ligation to other transcripts is reduced, allowing for specific sequencing of miRNAs and other 3′OH transcript species. In detail, 0.1–1 micrograms of total RNA per sample was used as the input material for the creation of small RNA libraries. Index codes were added to identify the sequences from each sample. Briefly, NEB 3′ SR adaptors were directly and specifically ligated to the 3′ ends of all miRNAs, siRNAs, and piRNAs. After the 3′ ligation reaction, the SR RT primer was hybridized to the excess 3′ SR adaptor that remained free after the 3′ ligation reaction, and the single-stranded DNA adaptor was transformed into a double-stranded DNA molecule. This step is important to prevent adaptor–dimer formation. Additionally, dsDNAs are not substrates for ligation mediated by T4 RNA Ligase 1 and, therefore, do not join with the 5′ SR adaptor in the subsequent ligation step. The 5′-end adapter was ligated to the 5′ ends of all miRNAs, siRNAs, and piRNAs. Then, first-strand cDNA was synthesized using M-MuLV Reverse Transcriptase (RNase H-). PCR amplification was performed using the LongAmp Taq 2× Master Mix, SR Primer from Illumina and an index (X) primer (New England Biolabs, Ipswich, MA, USA). The PCR products were purified on a 5% polyacrylamide gel (120 V for 60 min). DNA fragments corresponding to approximately 140–160 bp (the length of small non-coding RNA plus the 3′ and 5′ adaptors) were recovered and dissolved in 12 μL of TE buffer. Finally, the library quality was assessed on the Agilent Bioanalyzer 2100 system using DNA High-Sensitivity Chips (Agilent Technologies, St. Clara, CA, USA). The workflow is shown in Figure 6.

### 2.6. Clustering and Sequencing

The sequencing was carried out on an Illumina NextSeq550 platform (New England Biolabs (NEB), Ipswich, MA, USA), and approximately 75 bp paired-end (PE) raw reads were generated. At least 1–5 million reads per sample were generated.

### 2.7. Sequencing Data Analysis of Small RNA

The analysis process was automated using a customizable system developed in-house. First, the raw reads were pre-processed by FastQC [10] (Version 0.11.9, Andrews, Babraham Bioinformatics, England), MultiQC [11] (Version 1.12, Ewels et al., Stockholm, Sweden), Cutadapt [12] (Version 4.1, Marcel Martin, Dortmund, Germany), and BBMap [13] (Version 35.85, Brian Bushnell, Berkeley, CA (USA)), using the specific NGS technology configuration parameters. This pre-processing removed low-quality, ambiguous, and low-complexity stretches, linkers, adaptors, vector fragments, organelle DNA, polyA/polyT tails, and contaminated sequences, while keeping the longest informative part of the read. Subsequently, clean reads were aligned against the Human Genome Assembly (hg38) using STAR [14] (Version STAR-2.7.9a, Alexander Dobin, Harbor, NY, USA) to search matched reads. Expression levels were reported as CPM (counts per million). The miRNA reads were aligned to reference reads—in this case, the latest versions of miRNA databases (http://www.mirbase.org/) (accessed on 20 October 2022) [15,16]—using STAR 2.7.9a. The miRNA counts were included in a working matrix to carry out an analysis of differential expression between the study groups in R. In addition, a control threshold of the samples was established to eliminate miRNAs with high rates of missing data. After that, the different datasets were compared to identify genes showing differential expression. This process involved previous normalization of gene expression data and comparison of datasets using the edgeR library [17] (Version 3.40, Robinson et al., Victoria, Australia). Statistical tests and *p*-value fit measures were dependent on the type of data and the preliminary results [18]. We analyzed the remainder of the data using standard descriptive statistics (i.e., Student’s *t*-test, Fisher’s test, or the Mann–Whitney U test for comparison of groups). Comparison of more than two groups was performed by ANOVA or the Kruskal–Wallis test. Correlations between variables were assessed using Spearman’s test. Multiple linear regression analysis was performed to explore the presence or absence of collinearity among variables and to determine the validity of bivariate correlations. The level of rejection of the null hypothesis was 0.05. The result was a panel of genes or differently expressed miRNAs that met requirements such as stability across the population, minimum levels of expression, and statistical significance, among others. In the final stage, the predicted targeted genes were further analyzed by the Gene Ontology (GO) [19] (Version 4 December 2022 10.5281/zenodo.7407024, Ashburner et al., Indiana, USA), KEGG [20] (Version 104.1, Kanehisa Laboratories, Tokyo, Japan), and Reactome [21] (Version 83, Jassal et al., Toronto, ON, Canada) pathway databases for the visualization, interpretation, and analysis of known pathways in which they are involved.

### 2.8. Validation

To confirm the small RNA sequencing results, differentially expressed small RNAs need to be examined by qRT-PCR. If they turn out to be consistent with the small RNA sequencing results, the small RNA sequencing data are reliable. The discovered biomarker signature can therefore be assumed after data validation.

### 2.9. Biological Materials, Reagents, and Equipment

Primary human cardiac myocytes (HCMs) (PromoCell, Heidelberg, Germany, cat. no. C-12810)Myocyte Basal Medium (PromoCell, Heidelberg, Germany, cat. no. C-22270).Myocyte Growth Medium Supplement Mix (PromoCell, Heidelberg, Germany, cat. no. C-39275).Penicillin (5,000,000 Units, Sandoz, Basel, Switzerland).Streptomycin (1 g, Reig Jofré, Barcelona, Spain).DetachKit (PromoCell, Heidelberg, Germany, cat.no. 41210).Trypan Blue Solution, 0.4% (Lonza, Basel, Switzerland, cat. no. 17-942E).Standard salts and reagents for buffer preparation: HEPES, MES, NaCl, KCl, CaCl_2_, MgSO_4_, K_2_SO_4_, Na_2_SO_4_, MgCl_2_, D-glucose, NaOH, HCl (Lonza, Basel, Switzerland, cat. no. 17-516Q).miRCURY LNA™ miRNA Mimic HAS-MIR-197-3P (Qiagen, Hilden, Germany, cat. no. 339173YM00471956-ADA).miRNA Mimic Negative Control-FAM-5´conjugated (Qiagen, Hilden, Germany, cat. no. 339173YM00479902-ADB).miRCURY LNA™ miRNA Inhibitor Control (Qiagen, Hilden, Germany, cat. no. 339126YI00199006-ADB).TransIT-siQUEST Transfection Reagent (Mirus, Madison, WI, USA, cat. no. MIR 2114).RPMI 1640 basal medium (Gibco™, Waltham, MA, USA, cat. no. 22400089).RNase-free water (Lonza, Basel, Switzerland, cat. no. BE51200).Hank’s Balanced Salt Solution with Ca and Mg, *w*/*o* Phenol Red (Capricorn Scientific GmbH, Ebsdorfergrund, Germany, cat. no. BBSS-1A).Cytochalasin B (Cayman Chemical, Ann Arbor, MI, USA, cat. no. 11328-5).Sevoflurane (Sevorane ^®^, Sedana Medical, Danderyd, Sweden, AnaConDa Syringe).Propofol (5 amp–200 mg/20 mL. Injectable emulsion 1%. Fresenius Kabi-Austria GMBH, Graz, Austria)RNeasy Plus Universal Kit (Qiagen, Hilden, Germany).Qubit^®^ RNA Assay Kit (Invitrogen, Waltham, MA, USA).RNA Nano 6000 Assay Kit (Agilent Technologies, St. Clara, CA, USA).NEBNext^®^ Multiplex Small RNA Library Prep Set for Illumina^®^ (New England Biolabs (NEB), Ipswich, MA, USA).Water bath (Selecta-P. Mod Digiterm 100, Barcelona, Spain).CryoMed Controlled-Rate Freezers (Thermo Fisher, Waltham, MA, USA, Mod 802).Incubator (Eppendorf, Hamburg, Germany, mod. CellXpert-C170i).Thermocycler (Bio-Rad, Hercules, CA, USA).Nikon Eclipse TE2000-U Inverted Fluorescence Microscope (Nikon, Tokyo, Japan).Hypoxia chamber (BioSpherix, Parish, NY, USA).NanoPhotometer^®^ spectrophotometer (IMPLEN, Westlake Village, CA, USA).Qubit^®^ 3.0 Fluorometer (Life Technologies, Carlsbad, CA, USA).Agilent Bioanalyzer 2100 System (Agilent Technologies, St. Clara, CA, USA).Illumina NextSeq^®^ 550 System (New England Biolabs (NEB), Ipswich, MA, USA).Light Cycler 480 II de Roche (Roche, Indianapolis, IN, USA).PowerPacTM Basic Power Supply (Bio-Rad, Hercules, CA, USA).Molecular Imager^®^ ChemidocTM XRS+ Imaging System (Bio-Rad, Hercules, CA, USA).

### 2.10. Reagent Setup

Complete myocyte medium: Once the entire contents of the supplement are transferred into the basal medium, add 150 µL of streptomycin (0.1 mg/mL) and 200 µL of penicillin (100 UI/mL). Keep at 4 °C for up to 1 month.miRNA Mimic Negative Control-FAM-5´conjugated (66.7 µM): Add 75 µL of RNase-free water to 5 nmol of miRNA Mimic Negative Control to obtain a final solution concentration of 66.67 µM. Aliquot the final solution in volumes of 5 µL at −20 °C.miRNA Mimic Inhibitor Control: The inhibitor control does not need to be aliquoted.miRNA Mimic MIR-197-3P (66.67 µM): Add 75µL of RNase-free water to 5 nmol of miRNA Mimics Negative Control to obtain a final solution concentration of 66.67 µM. Aliquot the final solution in volumes of 5 µL at −20 °C.TransIT-siQUEST^®^ Transfection Reagent: Dilute 7.5 µL of TransIT-siQUEST^®^ Transfection Reagent in 250 µL of serum-free RPMI basal medium. Add 0.22 µL of miRNA Mimic Negative Control from the stock solution (final concentration of 5 nM per well)Cytochalasin B: Dissolve 5 mg of cytochalasin B and 250 µL of DMSO (dimethyl sulfoxide) as a stock solution (20 µg/µL). As described in [22], add 5 µg of cytochalasin B per mL of culture medium. Add 0.25 µL from the stock solution per mL of culture medium.

## 3. Discussion

Many biological processes are regulated by miRNAs [23], and their levels of expression are affected in many human diseases [24]. Currently, there is promising evidence that the use of miRNAs in clinical practice constitutes a reliable tool for future use, although there is still a lack of standardized protocols. These molecules meet most of the required criteria for being ideal biomarkers, such as high sensitivity, specificity, and accessibility. Despite their present limitations, miRNAs as biomarkers for various conditions remain an impressive research field. As current techniques evolve, we anticipate that miRNAs will become a routine tool in the development of personalized patient profiles, enabling more specific therapeutic interventions [25]. miRNAs have been proposed as an alternative to cell and gene therapies for cardiac regeneration [26,27]; however, these miRNA-based therapies still require clinical validation in models with greater translational potential [28,29]. This study was carried out with primary human cardiomyocytes without subjecting the patients to serious surgical risk, showing a methodological procedure that could be applied in the validation of the miRNAs identified as biomarkers in cardiac repair [30,31,32]. The differentially expressed miRNAs could be identified as biomarkers or therapeutic targets in cardiac protection induced by halogenated hypnotics.

The immediate clinical translation of the results derived from our proposed in vitro protocol is clearly justified by the clinical application of the identified miRNAs as biomarkers of cardiac protection.

## Figures and Tables

**Figure 1 life-13-00064-f001:**
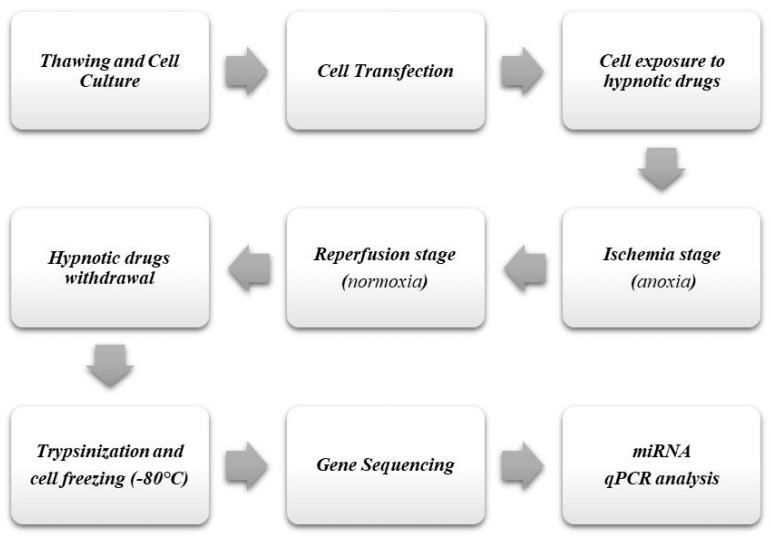
Schematic representation of the complete experimental design.

**Figure 2 life-13-00064-f002:**
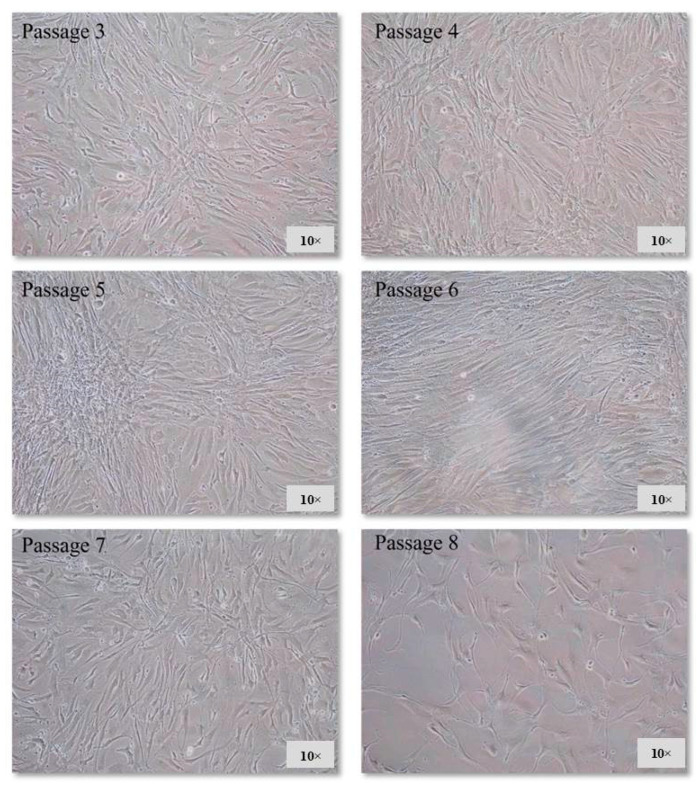
Images of HCM culture passages acquired on the seventh day of culture.

**Figure 3 life-13-00064-f003:**
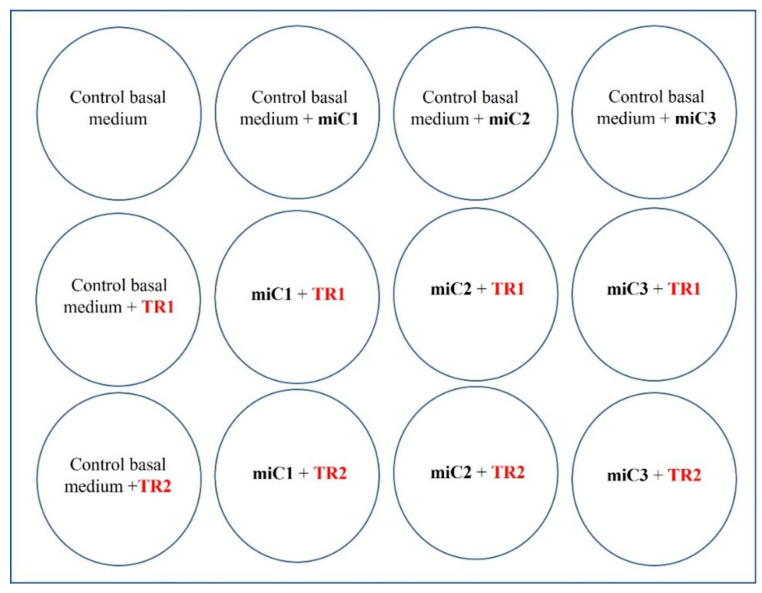
Culture plate design to perform the transfection protocol: Concentration of miRNA Mimic Negative Control-FAM-5′conjugated (66.67 µM) added to HCM culture per well: miC1 = 2.5 nM~5 pmol (0.07 µL/mL), miC2 = 5 nM~10 pmol (0.14 µL/mL), and miC3 = 10 nM~50 pmol (0.7 µL/mL). Concentration of TransIT-siQUEST^®^ transfection reagent added per well: TR1 = 3 µL, TR2 = 5 µL. Final volume per well: 1 mL. Final number of HCMs per well: 57 × 10^3^ cells.

**Figure 4 life-13-00064-f004:**
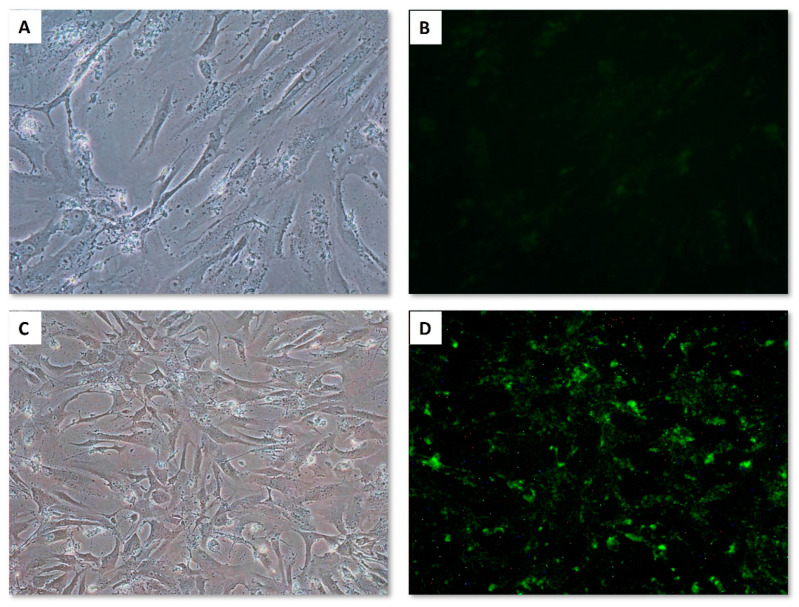
Primary human cardiomyocytes 24 h post-transfection: (**A**) non-transfected HCMs, 10× optical microscopy objective; (**B**) non-transfected HCMs, 10× fluorescence microscopy image; (**C**) transfected HCMs, 10× optical microscopy objective; (**D**) transfected HCMs, 10× fluorescence microscopy objective. HCMs were transfected with 3 µL of TransIT-siQUEST^®^ transfection reagent and miRNA Mimic Negative Control-FAM, to a final concentration of 5 nM.

**Figure 5 life-13-00064-f005:**
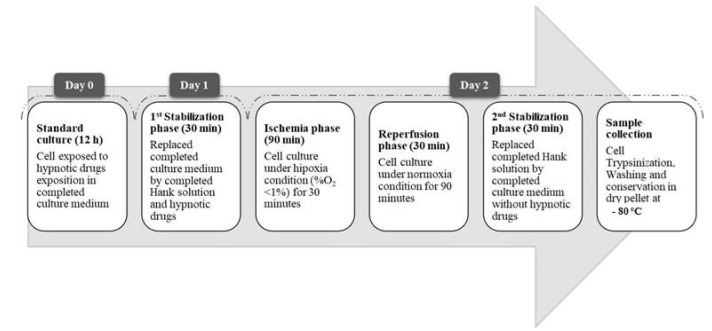
Ischemia/reperfusion: in vitro procedure workflow.

**Figure 6 life-13-00064-f006:**
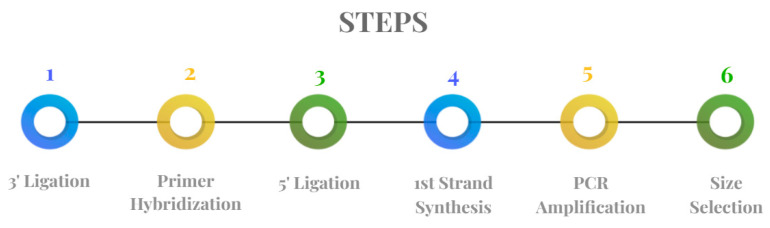
Multiplex small RNA library prep workflow.

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
