# Peer review of "Identification of miRNAs as Biomarkers of Cardiac Protection in Non-Genetically Modified Primary Human Cardiomyocytes Exposed to Halogenated Hypnotics in an In Vitro Model of Transfection and Ischemia/Reperfusion: A New Model in Translational Anesthesia"

_life, 2022, doi:10.3390/life13010064_

Round 1

Reviewer 1 Report

The current Manuscript submitted by Maria Dolores C L et al and entitled as “Identification of miRNAs as biomarkers of cardiac 2 protection in non-genetically modified primary human 3 cardiomyocytes exposed to halogenated hypnotics in an in vitro 4 model of transfection and ischemia/reperfusion: A new model 5 in translational anesthesia”. Manuscript hypothesized and discussed mainly on identification of cardio protection miRNAs as a biomarkers during the anesthetic condition in an in vitro primary cell culture (Cardiomyocytes) model under the condition of ischemic/reperfusion which mimic the myocardial conditioning procedure. However, several concerns need to be addressed.

Major concerns: 

1) Authors found differential expression of miRNA in HCM exposed to different hypnotic drugs and predicted the functional targets of miRNAs.  However, it would be more evident and add-up to this current study if authors can validate a couple of major miRNAs identified in this study and show the cardio protective effect of those miRNAs using current in vitro model by over expressing the specific miRNAs and their regulated target signaling proteins. 

2) It is still not clear that how halogenated anethetics can regulate the miRNA expression and is there any evidence of specific cardio protective miRNAs and the miRNAs that contribute to disease progression targeted by specific halogenated compounds tested in this study? List the miRNA is such found. 

3) As authors found cardio protective miRNA up on administration of halogenated anethetics, it would be more valuable to this current study if authors can show the cardio protective effect of identified miRNA in current in vitro model by silencing the specific miRNA under the administration of halogenated drugs and their target signaling proteins.

Minor:

1)  Line 87: authors said cardiac cells. Please specify of which cardiac cells derived from human heart tissue? 

2)  In figure 1, In schematic diagram please indicate the exposure of hypnotic drugs with-in the check Box named 1st stabilization. This will allow readers to fallow the procedure.

3) Line 95 to line 108, Authors described complete procedure of experimental design. However, the description seems to be not matching with the representative Figure 1. especially, the terminology of Stabilization is very confusing and 2nd stabilization is a control conditions without hypnotic drugs. However, in the schematic diagram appearing as entire experimental procedure, nothing to do with 1st and 2nd stabilization as it was added. Therefore, just delete the stabilization step and replaced the Box as presence or absence of hypnotic drugs

Reviewer 2 Report

The article is important, and tries to find a biomarker that can be used in routine in the clinical practice. However, there are some minor points that must be considered:

1 - It is important to consider the parameters to be considered a good biomarker (selectivity, price of performing the procedure, ....) discuss the importance of this biomarker.

2- The cells used in this procedure are a cell line? what is the reference of the cells? Characterize these cells better.

3- The images are very important and make a lot of sense, however, the quality could be better.

4- The discussion should be improved, analyze the importance of miRNAs, the characteristics of a biomarker, the selectivity that seems to be low of this marker, the clinical importance of its realization....

Round 2

Reviewer 1 Report

This is the revised version of Manuscript. All the comments were addressed properly and changes were made accordingly. Current version can be accepted for publication. 

Author Response

Thank you for your comments

Reviewer 2 Report

This article was revised appropriately.

I recommend accept

Author Response

Thank you for your comments